# Environmental DNA concentrations are correlated with regional biomass of Atlantic cod in oceanic waters

Ian Salter [1]*, Mourits Joensen[1], Regin Kristiansen[1], Petur Steingrund[1] & Poul Vestergaard[1]

Environmental DNA (eDNA) has emerged as a powerful approach for studying marine fisheries and has the potential to negate some of the drawbacks of trawl surveys. However, successful applications in oceanic waters have to date been largely focused on qualitative descriptions of species inventories. Here we conducted a quantitative eDNA survey of Atlantic cod (*Gadus morhua*) in oceanic waters and compared it with results obtained from a standardized demersal trawl survey. Detection of eDNA originating from Atlantic cod was highly concordant (80%) with trawl catches. We observed significantly positive correlations between the regional integrals of Atlantic cod biomass (kg) and eDNA quantities (copies) ($R^2 = 0.79$, $P = 0.003$) and between sampling effort-normalised Catch Per Unit Effort (kg hr$^{-1}$) and eDNA concentrations (copies L$^{-1}$) ($R^2 = 0.71$, $P = 0.008$). These findings extend the potential application of environmental DNA to regional biomass assessments of commercially important fish stocks in the ocean.

[1] Faroese Marine Research Institute, Nóatún 1, Tórshavn FO-100, Faroe Islands. *email: ians@hav.fo

The successful management of commercial fisheries relies on standardised surveys to estimate the quantity and distribution of fish stocks. Atlantic cod (*Gadus morhua*) is an iconic example that demonstrates how poorly constrained data and uninformed decision making can result in catastrophic stock decline and ensuing economic and social problems[1]. Traditional stock assessments of demersal fish species have relied primarily on trawl surveys, which have provided a valuable stream of information to decision makers[2]. However, there are some notable drawbacks of demersal trawl surveys including cost[3], gear selectivity/catchability[4], habitat destruction[5] and restricted coverage (e.g. hard-substrate bottom environments, marine protected areas).

Environmental DNA (eDNA) has emerged as a potentially powerful alternative for studying ecosystem dynamics. The constant loss and shedding of genetic material from macrogranisms imparts a molecular footprint in environmental samples that can be analysed to determine either the presence of specific target species[6,7] or characterise biodiversity[8,9]. The combination of next generation sequencing and eDNA sampling has been successfully applied in aquatic systems to document spatial and temporal patterns in the diversity of fish fauna[10–13]. To further develop the utility of eDNA for fisheries management, understanding the ability of eDNA quantities to reflect fish biomass in the ocean is an important next step.

Positive relationships between eDNA quantities and fish biomass and abundance have been demonstrated in experimental systems[14–16]. However, known variations between eDNA production[17,18] and degradation[19–22] rates is anticipated to complicate these relationships in natural systems. Furthermore, in oceanic systems, large habitat volumes and strong currents are likely to result in physical dispersal of DNA fragments away from target organisms[23]. These confounding factors have been previously considered to restrict the application of quantitative eDNA monitoring in oceanic settings[24].

Despite these potential constraints, numerous studies in marine environments have found positive relationships between eDNA quantities and complimentary survey efforts including radio-tagging[25], visual surveys[13,26], echo-sounding[27] and trawl surveys[12,28]. However, studies that quantify target eDNA concentrations of commercial fish species with standardised trawl surveys in marine environments are much scarcer[28]. In this context, direct comparisons of eDNA concentrations with biomass and stock assessment metrics, such as Catch Per Unit Effort (CPUE), are necessary to understand the applicability of eDNA monitoring to contribute to fisheries management efforts.

The present study focuses on the application of quantitative eDNA monitoring targeting Atlantic cod because it is a species of commercial importance that has been subjected to strong fishing pressure. Atlantic cod is a valuable commodity for the Faroese fishing economy, accounting for an export value of ~0.9 billion Danish Krona in 2018 (Faroese National Statistics Office). Sustained demersal surveys between 1994 and 2018 have shown a notable decline in Atlantic cod biomass in Faroese waters[2], particularly in the Faroe Bank region. Fisheries management closed the Faroe Bank to all fishing gears in 2009, except for minor jigging during summertime. Understanding the recovery and spatial distribution of Atlantic cod is extremely relevant for the Faroese fishing economy. Here we used qPCR detection to measure the concentrations of Atlantic cod eDNA in bottom water samples around the Faroe Islands and compared this with biomass estimates obtained from a parallel standardised demersal survey. We apply these data to test the hypothesis that there is a positive correlation between the regional biomass of Atlantic cod and eDNA concentrations in oceanic waters.

## Results

**Specificity and validation of Atlantic cod qPCR assay.** To test the specificity of the commercial qPCR (Techne, Bibby Scientific, United Kingdom) primers for *G. adus morhua* (Atlantic cod) we performed cross-amplification tests on related non-target species collected from the survey area (Table 1; Supplementary Table 1). Analysis of the survey trawl data revealed eight species belonging to the Gadidae family. Haddock (*Melanogrammus aeglefinus*) and saithe (*Pollachius virens*) were present at biomass levels one order of magnitude lower than Atlantic cod. Norway pout (*Trisopterus esmarkii*), whiting (*Merlangius merlangus*) and blue whiting (*Micromesistius poutassou*) were present at two orders of magnitude lower and poor cod (*Trisopterus minutus*) and silvery pout (*Gadiculus argenteus thori*) at three orders of magnitude lower. DNA extracted from tissue specimens amplified for Atlantic cod at a Cq value of 18.8 ± 0.56, no amplification was observed for non-target species. Due to the low abundance of poor cod it was not possible to collect a specimen for cross-amplification tests. However, considering it's negligible biomass and regional distribution, compared to that of Atlantic cod (Supplementary Table 1), we assume zero interference. The qPCR assay was thus deemed specific for Atlantic cod in Faroese Waters.

Filtration, DNA extraction and qPCR amplification of Atlantic cod eDNA was tested in-vitro from small volume (1.5 L) water samples collected at the Faroese national aquarium (Føroya Sjósavn). DNA extracted from water samples of three separate tank systems, each containing Atlantic cod, successfully amplified with the *G. morhua* qPCR primer (Supplementary Fig. 1). Limits of detection (LOD) and limits of quantification (LOQ) for the quantitative PCR assay were determined from the analysis of a 10-point standard replicate curve (Supplementary Figs. 2 and 3). LOD and LOQ were 3 and 20 copies per reaction, corresponding to an LOD of 48 and LOQ of 320 copies per litre.

**Regional analysis of historical trawl data.** Analysis of historical trawl data (1994–2018) show a strong regional distribution of Atlantic cod around the Faroe Islands (Fig. 1, Supplementary Fig. 4). The standardised demersal survey has occupied the same grid positions over its 25-year history, allowing us to analyse historical catch data from the exact sampling positions of the present study (Fig. 1b, d, Supplementary Table 2). High biomass area are typically located in the regions North, West and Bank central of the Faroe Islands with median (and maximum) CPUEs of 1566 (5187), 703 (7080) and 98 (2689) kg h$^{-1}$, respectively. Regions of intermediate biomass are the East coast (<150 m), South and East shelf (<150–200 m), with CPUEs of 91.4 (626), 27.5 (92.3) and 11.0 (55.9) kg h$^{-1}$, respectively. Low biomass regions were identified on the edge of the Faroe Bank (Bank edge; median CPUE 5.37 kg h$^{-1}$) and further to the East, between 200 and 500 m (East deep; median CPUE 3.48 kg h$^{-1}$). Despite temporal fluctuations in total catches, this regional distribution of Atlantic cod has been quite consistent over the trawl-survey period (Fig. 1d, Supplementary Fig. 4).

**Detection of Atlantic cod by eDNA and trawl survey methods.** During the 2018 demersal survey a total of 35 paired trawl-eDNA sampling stations (Supplementary Table 3) targeted eight distinct regions classified on the basis of historical trawl survey data and bathymetry (Fig. 1a). Catch per Unit Effort (CPUE) of Atlantic cod covered three orders of magnitude with a median of 107 and arithmetic mean of 697 kg h$^{-1}$. Consistent with the historical distribution, highest CPUE's were found in the West (max = 8312 kg h$^{-1}$) and North (max = 1531 kg h$^{-1}$). Negligible CPUE's were recorded in the East deep region (<3 kg h$^{-1}$), the

**Table 1 qPCR Specificity tests of the _Gadus morhua_ primer used for detecting Atlantic cod.**

| Common name | Scientific name | Total biomass (x $10^3$ kg) | Ratio with cod | Tissue type | Cq |
|---|---|---|---|---|---|
| **Atlantic cod** | _Gadus morhua_ | 18.8 | 1.00 | Muscle | 18.8 ± 0.56 |
| **haddock** | _Melanogrammus aeglefinus_ | 12.9 | 0.69 | Muscle | Undetermined |
| **saithe** | _Pollachius virens_ | 1.79 | 0.10 | Muscle | Undetermined |
| **Norway pout** | _Trisopterus esmarkii_ | 0.88 | 0.05 | Muscle | Undetermined |
| **whiting** | _Merlangius merlangus_ | 0.58 | 0.03 | Muscle | Undetermined |
| **blue whiting** | _Micromesistius poutassou_ | 0.31 | 0.02 | Fin | Undetermined |
| **poor cod** | _Trisopterus minutus_ | 0.03 | 0.002 | No sample | No sample |
| **silvery pout** | _Gadiculus argenteus thori_ | 0.01 | 0.001 | Muscle | Undetermined |

Total biomass is the sum of all bottom trawl catches in the study area. Ratio with cod is the biomass of the related Gadidae species relative to Atlantic cod. Cq is the cycle quantification value from qPCR. Undetermined means the threshold was not exceeded after 50 cycles. Due to the extremely low abundance in the study area it was not possible to collect a tissue sample for poor cod

East Shelf region ($<10\,\mathrm{kg\,h^{-1}}$) and Bank Edge ($<8\,\mathrm{kg\,h^{-1}}$) (Fig. 1b).

Considering the paired trawl-eDNA stations, Atlantic cod was caught at 27 trawling locations (77%). The spatial distribution of detection rates is shown for the trawl (Fig. 2a) and eDNA surveys (Fig. 2b). Four out of seven of the failed eDNA detections corresponded to trawls exhibiting catch rates < $10\,\mathrm{kg\,h^{-1}}$; excluding these negligible trawl catches establishes concordance between detection methods at 80%. Less than 6% (2/35) of sampling locations exhibited positive eDNA detection discordant with positive trawl detection. To assess association between trawl and eDNA detection methods we calculated the mean square contingency coefficient (Φ). Considering all data, Φ was calculated at 0.48, increasing to 0.62 if negligible trawl catches ($<10\,\mathrm{kg\,h^{-1}}$) are classified as negative trawl detections.

**Concentrations of Atlantic cod eDNA in the survey area.** Atlantic cod eDNA copy numbers were measured above the LOQ at 21 out of the 35 survey stations (Supplementary Table 4). Highest copy numbers were recorded in the West region (71,136 copies $\mathrm{L^{-1}}$), followed by the North region (49,337 copies $\mathrm{L^{-1}}$). None of the samples from the East deep and Bank edge region displayed positive amplification (Fig. 1c). None of the field sampling or extraction blanks displayed amplification, ruling out possible errors resulting from contamination.

To address the quantitative relationships between catch data and eDNA copy number we summed the paired trawl-eDNA station values within the pre-defined regions (Fig. 1a) to calculate regional integrals. Mapping the regional distribution of Atlantic cod biomass inferred from the trawl survey against the eDNA survey showed good spatial correspondence (Fig. 2c, d) as evident from the regression coefficient of the regional ranks ($R^2 = 0.64$, $P = 0.0102$). The highest biomass of $14.4 \times 10^3$ kg per region was found in the West together with the highest concentration of Atlantic cod-derived eDNA ($151.2 \times 10^3$ copies per region). No eDNA was detected in the East deep and Bank edge regions, corresponding to negligible trawl biomass of <10 kg per region. Assuming a median size of 3.24 kg individual$^{-1}$ (Supplementary Fig. 5), 10 kg is equivalent to approximately three individuals. Regional integrals of biomass and DNA copy numbers were ranked 1–8 in descending order and plotted against each other (Supplementary Table 5; Fig. 3a). Complete agreement in these rankings would result in all data falling on the 1:1 line. Rankings displayed zero residual variance at the high and low end of the regional biomass gradient.

**Comparison of eDNA concentrations and trawl biomass.** Trawl-derived biomass estimates of Atlantic cod were characterised by logarithmic normal distributions (Supplementary Fig. 6) and so logarithmic transformation was applied to regional

biomass integrals to obtain a normally distributed dataset (Supplementary Fig. 7) for regression analysis. We used the Shapiro-Wilk test to validate that both dependent and independent variables were normally distributed prior to regression modelling. All variables and residuals used in the regional regression models were characterised by Shapiro–Wilk p-values > 0.05 confirming normal distributions. Prior to regression we checked for spatial autocorrelation amongst variables with Moran's I, employing an inverse distance matrix. Spatial autocorrelation was ruled out for all regression variables at a significance level of 95%.

To test our central hypothesis we evaluated the relationship between eDNA concentrations and trawl-biomass of Atlantic cod in the standardised survey region (Fig. 3). We detected a significantly positive correlation between the regional integrals of cod biomass (kg) and eDNA concentrations (copies) (Type II regression, $R^2 = 0.79$, $P = 0.003$, $y = 37709 \times -34009$, Fig. 3b). The correlation between Catch per Unit Effort (CPUE) of Atlantic cod ($\mathrm{kg\,h^{-1}}$) and (copies per litre) was significantly positive with a smaller regression coefficient (Type II regression, $R^2 = 0.71$, $P = 0.008$, $y = 10016 \times -3540$, Fig. 3c). The positive correlation between eDNA concentrations and CPUE on a station-by station basis was statistically significant but considerably weaker than the regional comparisons (Type II regression, $R^2 = 0.18$, $P = 0.01$, $y = -12910 \times -7998$, Fig. 3d). Excluding eDNA concentration data equal to zero results in a statistically insignificant correlation (Type II regression, $R^2 = 0.02$, $P = 0.5$) on a station-by-station basis.

**Discussion**
The objective of the present study was to test the hypothesis that regional variability in eDNA concentrations are quantitatively correlated with regional biomass of Atlantic cod in oceanic waters. To correctly address this objective it was necessary to sample over a representative biomass gradient (unknown at the time of sampling) characterising an ~100,000 $\mathrm{km}^2$ standardised survey area in Faroese waters. On the basis of historical (25 years) CPUE data obtained from standardised trawl surveys we were able to identify eight distinct regions that were targeted for a quantitative eDNA survey (Fig. 1). The Trawl CPUE data obtained from the 2018 eDNA-trawl survey followed the historical regional distributions permitting us to test our main hypothesis.

Concentrations of Atlantic cod eDNA in bottom waters in the survey area exhibited significant regional variability. We observed a range of 0–$10^4$ copies $\mathrm{L^{-1}}$ of Atlantic cod eDNA in Faroese waters, comparable to the 0–$10^4$ copies $\mathrm{L^{-1}}$ recently reported in the Baltic Sea[28]. There have been few studies that have attempted to apply qPCR-based quantification of target-species eDNA concentrations in open sea systems[26–29]. Whale shark and mackerel tuna eDNA concentrations have been shown to vary from 0 to $10^6$ and $10^2$ to $10^8$ copies $\mathrm{L^{-1}}$, respectively, in the

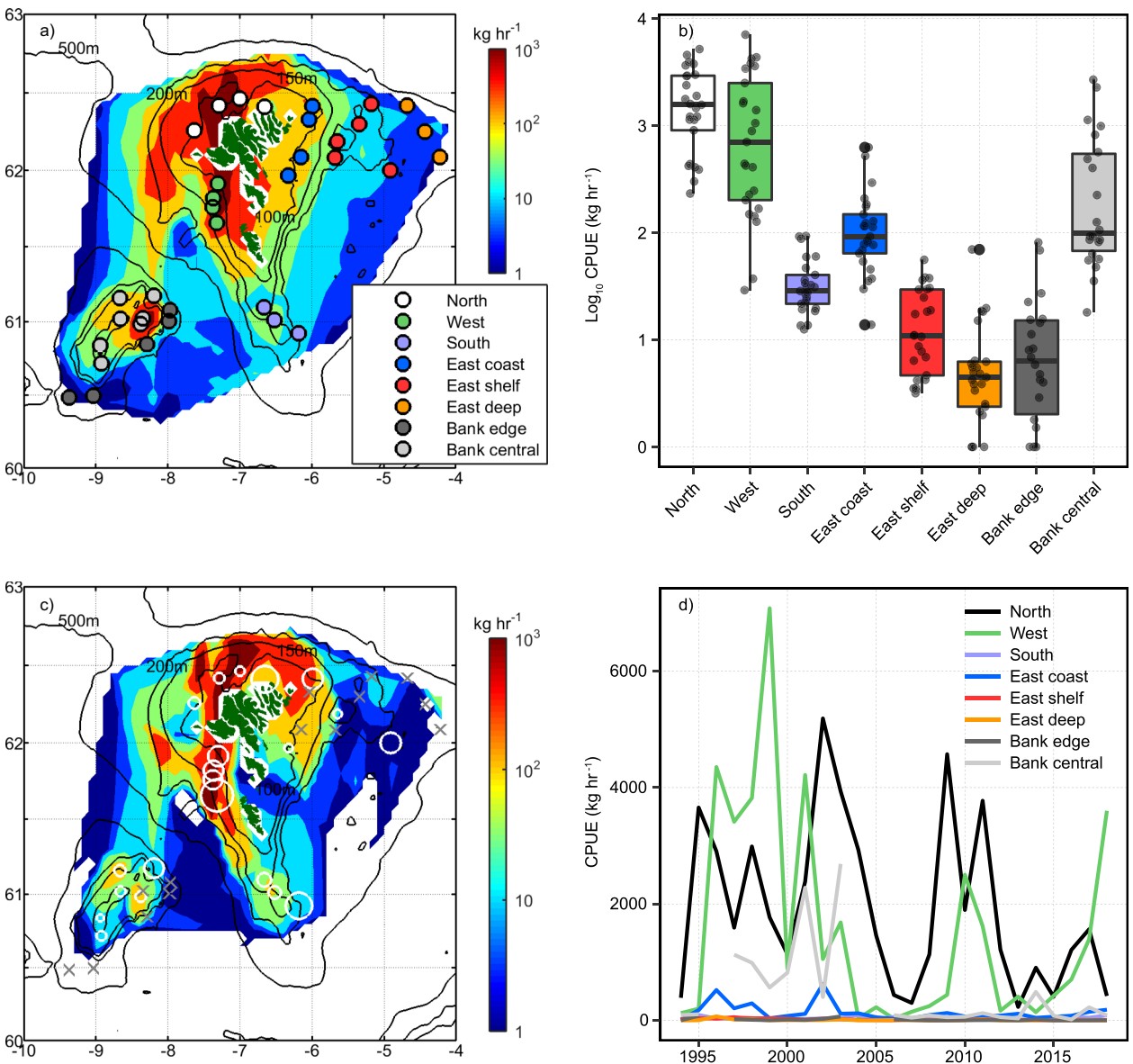

**Fig. 1** Regional analysis of Catch Per Unit Effort (CPUE) for Atlantic Cod. Panel (**a**) shows the sampling area for spring demersal trawl survey of the Faroese Marine Research Institute. Filled contours are interpolated data points of Catch Per Unit Effort (CPUE: kg h$^{-1}$) for Atlantic cod for the entire survey period (1994–2017; $n = 3090$). Filled circles identify the sampling positions of the paired eDNA water and trawl survey. Colours of the filled circles correspond to region assignment based on bathymetry and historical CPUE data. Panel (**b**) Box and whisker plot showing the statistical distribution of historical CPUE data (1994-2017) for different regions. The data have been log-transformed. The median (Q2) is used to describe central tendency and is denoted by the horizontal bar. The upper and lower hinges represent the 75% (Q3) and 25% (Q1) percentiles, respectively. Interquartile range (IQR) is defined as the difference between the 75 and 25% percentiles (Q3-Q1). The lower whisker is the smallest observation $\geq$ Q1-(1.5*IQR). The upper whisker is the largest observation $\leq$ Q3 + (1.5*IQR). Individual data points are displayed as points. A statistical summary of non-transformed data, including population size ($n$) is provided in Supplementary Table 2. Panel (**c**) shows CPUE data for the 2018 spring demersal survey. Filled contours are interpolated data points ($n = 128$). Open white circles are eDNA concentrations linearly scaled to the largest eDNA concentration of 71,136 copies L$^{-1}$. Grey crosses mark sampling positions where Atlantic cod DNA was not detectable by qPCR (no amplification). Panel (**d**) shows the inter-annual record of CPUE within each region, based on the fixed survey stations sampled in 2018 displayed in panel **a**. Total catches (kg) within a region were normalised to sampling effort (time) to determine annual values of CPUE (kg h$^{-1}$) for each survey year during the period 1994–2017.

Arabian Gulf[29]. Studies that have aimed to compare eDNA concentrations with alternative survey methods, including echosounding[27] and visual observations[26,29], have found a good degree of spatial concordance between the methods.

In the present study we observed significantly positive correlations between DNA concentrations and biomass of Atlantic cod, both in terms of regional integrals (kg per region and copies per region, Fig. 1b) and sampling effort-normalised values between regions (kg h$^{-1}$ and copies L$^{-1}$, Fig. 1c). Similar to

observations in the Baltic[28], we observed statistically weak correlations between CPUE and eDNA concentrations when assessed on a station-by-station basis (Fig. 1d). However, our study was purposefully designed to avoid such comparisons, which might be considered as counter-intuitive and uninformative in oceanic settings. Reference biomass measurements, in this case trawl survey data, typically exhibit very large variability over short spatial scales, particularly in the case of aggregating species like cod[30]. Furthermore, lagrangian modelling studies indicate that

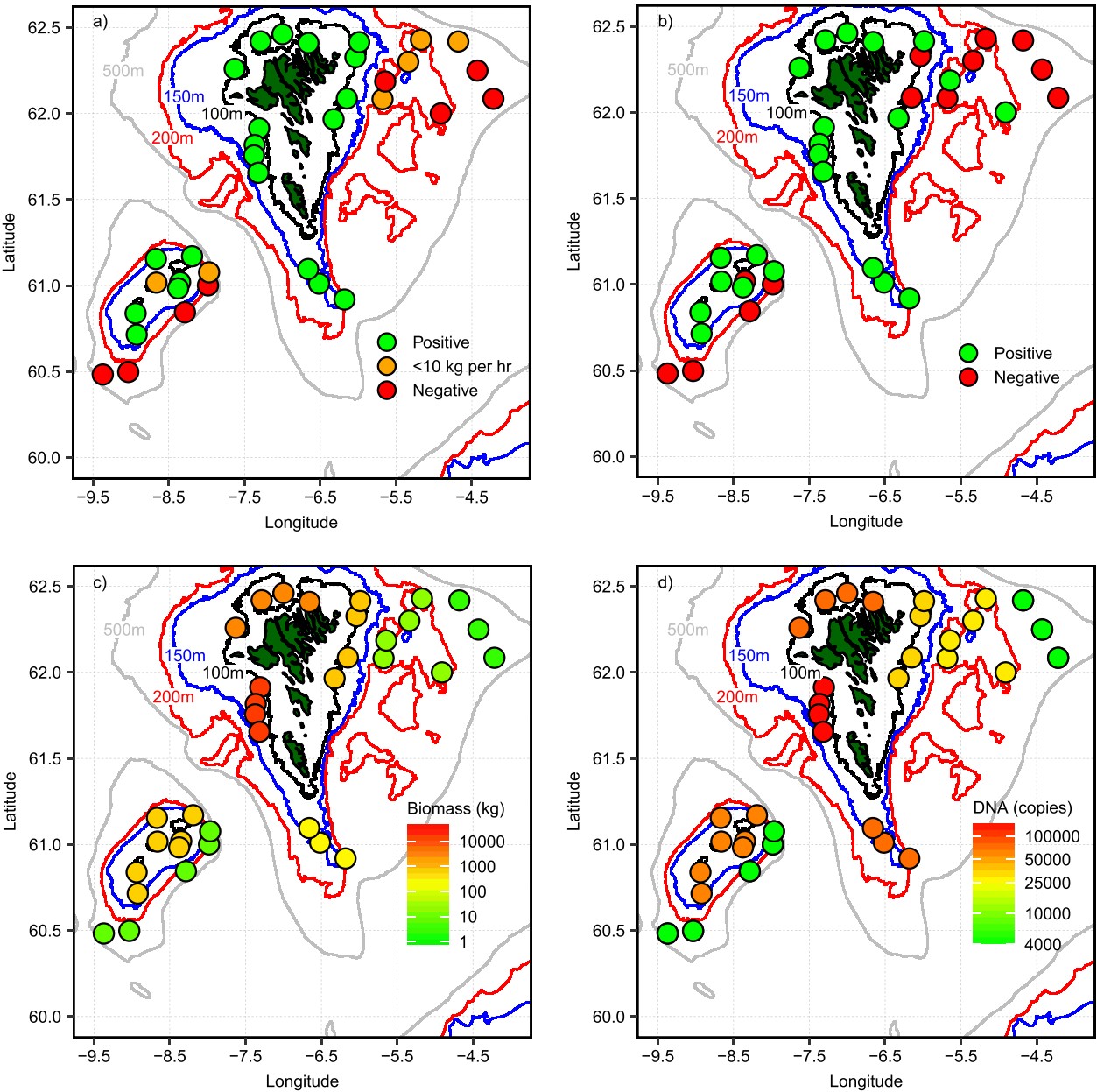

**Fig. 2** Regional detection rates and quantities from demersal trawl and eDNA survey. Panel (**a**) shows the detection of Atlantic cod from demersal trawl survey. Green and red filled circles denote positive and negative detection, respectively. Orange filled circles show positive detection at a biomass of <10 kg h$^{-1}$. Panel (**b**) shows positive and negative detections of Atlantic cod from environmental DNA samples. Green and red filled circles show positive and negative detections, respectively. Panel (**c**) shows region quantities of Atlantic cod from demersal trawl survey. Values represent the sum of biomass within each region; data is expressed logarithmically. Panel (**d**) shows region quantities of cod eDNA copies. Regional values are also provided in Supplementary Table 5.

both dilution and degradation kinetics decrease the probability of eDNA detection rates from the point of origin[23]. We observe a more robust statistical association between trawl and eDNA survey data for detection rates than for quantitative relationships on a station-by station basis (Phi-coefficient = 0.62 versus Pearson correlation coefficient of 0.18). It is worth noting that even this weak regression coefficient characterising the station-by-station comparison could be potentially biased due to a skewed distribution of eDNA copy numbers that result from numerous non-zero values for eDNA copies. Removing non-zero eDNA copy numbers results in an insignificant correlation on a station-by-station basis. Thus it appears that at a fine scale, eDNA signals are diluted and degraded to a level that confounds quantitative

relationships whilst still remaining amplifiable, and thus amenable to species detection by qPCR and next generation sequencing approaches.

These findings support our initial presumption of a regional approach and that quantitative relationships between eDNA and trawl-surveys are weak at fine-scale resolution in the ocean. For management purposes, stock estimates often rely on integrating multiple trawl catches over a larger spatial area[2] and we thus applied the same rational to our eDNA sampling for comparative purposes. Our data highlight the importance of spatial observing scales with respect to quantitative eDNA monitoring in oceanic settings. It is likely that the correct spatial scale for eDNA monitoring depends on the target species in question, variability

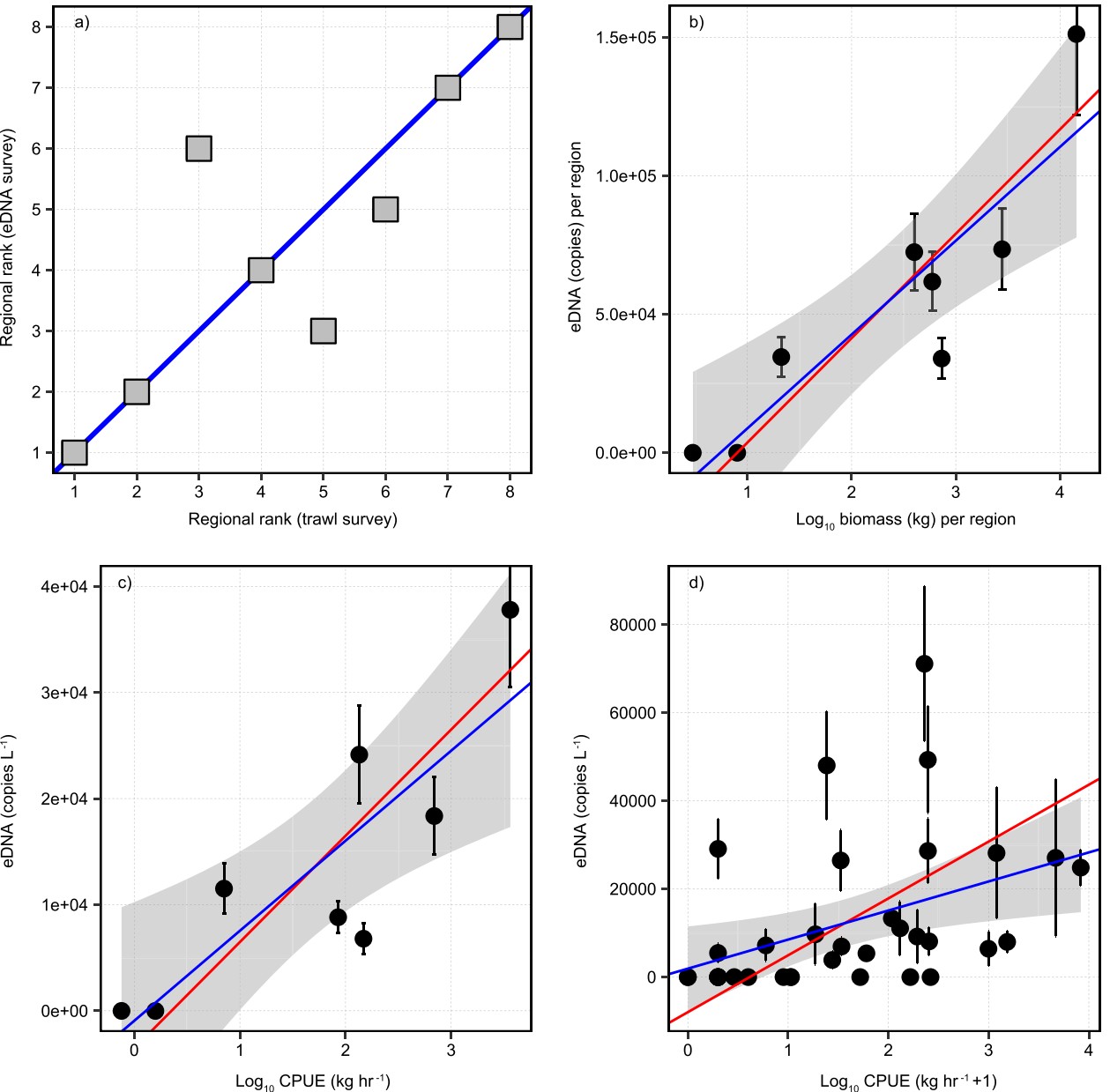

**Fig. 3** Quantitative relationships between trawl biomass and eDNA copy numbers of Atlantic cod. Panel (**a**) shows regional rank correlation. Regions of highest biomass were ranked in descending order and plotted against regions of highest eDNA copy numbers ($R^2 = 0.64$, $P = 0.0102$). The blue solid line denotes 1:1 relationship. In panels (**b–d**) the solid blue line denotes a type I (Ordinary Least Squares; OLS) regression model and the red line a type II (Ranged Major Axis; RMA) regression model. Shaded grey area is the 95% confidence interval calculated from type I regression. Type I regression assumes the biomass determined from the trawl survey as an independent and true reference value and thus regression error is associated only with eDNA concentrations. Type II regression takes into consideration that both trawl determined biomass and eDNA concentrations are field variables each containing variance (Legendre et al. 2018)[44]. Panel (**b**) shows the correlation between regional sums of Atlantic cod biomass obtained from trawl surveys (kg) and eDNA quantities (copies). Type I OLS model ($y = 33{,}888 \times - 25{,}416$; $r^2 = 0.76$; $p = 0.003$). Type II RMA model ($y = 37{,}709 \times - 34{,}009$; $r^2 = 0.79$, $p = 0.003$). Error bars on $y$-axis are propagated errors for region sums determined from the analysis of technical replicates and represent ± 1 sd. Panel (**c**) shows the correlation between CPUE for trawl survey (kg h⁻¹) and sampling effort normalised eDNA quantities (copies L⁻¹). Type I OLS model ($y = 8487 \times -948$; $r^2 = 0.66$; $p = 0.008$). Type II RMA model ($y = 10016 \times - 3540$; $r^2 = 0.71$, $p = 0.008$). Error bars on y-axis are propagated errors for copies L⁻¹ determined from the analysis of technical replicates and represent ± 1 sd. Panel (**d**) shows the correlation between CPUE for trawl survey (kg + 1 h⁻¹) and eDNA concentrations (copies L⁻¹) for individual stations. Type I OLS model ($y = 6596 \times + 1890$; $r^2 = 0.15$; $p = 0.01$). Type II RMA model ($y = 12910 \times - 7998$; $r^2 = 0.18$, $p = 0.01$).

in production[18,31] and degradation rates[22,32], local biogeochemical conditions[20,33], and local hydrography influencing signal dispersal and dilution[23]. In the present study, integrating eDNA and trawl observations over relatively few sampling points per region (<7) was sufficient to establish statistically robust regional relationships between eDNA and CPUE for Atlantic cod.

The Faroe bank region has exhibited significant temporal variability in stocks of Atlantic cod over the last 25 years (Supplementary Fig. 4) that have been linked to over-exploitation and led to a complete closure of the area to commercial fishing. Integrated over the same observational scale, multi-decadal trends in CPUE data from the Bank Central

region are in a similar range as the regional variation we report. If the linear dynamic range between biomass and eDNA copy number we derived from our spatial analysis are assumed to be valid as a function of time, our results suggest that a quantitative eDNA survey of similar spatial scale to the one described in this study would have correctly resolved the pattern of Atlantic cod stock decline on the Faroe Bank (Supplementary Fig. 8). Clearly, sustained long-term observations of both quantitative eDNA measurements and standardised demersal trawling surveys are required to fully validate the potential of eDNA monitoring to track stock fluctuations.

eDNA has emerged as a valuable monitoring tool for fisheries science and management[24]. Recent work has revealed its capacity to describe fish diversity[10,12], presence of endangered[34] and invasive species[25], predator-prey co-occurrence[29] and population genetics[29,35]. However, the comparison of quantitative eDNA with standardised survey metrics used in fisheries management are much rarer[28]. Here we show for Atlantic cod that quantitative eDNA monitoring displays positive correlations with CPUE data over regional scales in oceanic waters around the Faroe Islands. Increasing the spatial resolution and replication of sampling are likely to resolve these relationships further. The range of CPUE estimates correlated with DNA concentrations are comparable to the temporal decline of Atlantic cod (linked to overfishing) on the Faroe Bank. Quantitative eDNA monitoring may thus have the sensitivity to track regional stock fluctuations in the ocean. Our findings extend the potential application of eDNA to regional biomass assessments of commercial fish stocks in the ocean.

## Methods

**Standardised trawl survey.** Demersal trawling was carried out on board Magnus Heinason (Expedition numbers: 1806, 1808, 1810) as part of the standardised spring demersal survey conducted by the Faroese Marine Research Institute. During a five-week period from February–March 2018, 128 one-hour bottom-trawls were carried out on the Faroe Plateau and Faroe Bank. Trawl doors were of the Thyborøn type, and the bridle length was either 60 m (<140 m) or 120 m (>140 m). The length from the codend to the headrope was 40 m, and the distance between doors was either 70 m (60-m bridles) or 130 m (120-m bridles). Trawling gear was a 116-ft box trawl with a mesh-size of 135 mm. The headrope was located 4.5–5 m above the bottom. A net of 40-mm mesh and 8 m long was placed inside the codend and tows were conducted by day. The survey catch was sorted by species. The total weight of the survey catch and of the sorted species was recorded. Individuals of larger specimens were collected for weight and length measurements.

**eDNA sample collection.** Seawater samples of 1.5 L were collected at 35 of the trawling positions (Fig. 1a) from Niskin bottles mounted on a stainless steel CTD frame. The aim was to collect water samples 4 metres above the seafloor in order to correspond to the trawl height of 5 m and to minimise the possibility of sampling eDNA that might originate from non-recent sedimentary sources. However, due to occasional high sea states, and the significant pitch and roll of the research vessel, actual sampling depths ranged between 1.4 and 8.7 m.a.b with a median of 3.9 m.a.b. Water samples were collected immediately prior to the trawl to minimise contamination from trawl-derived DNA sources.

**Onboard processing and contamination controls.** Careful measures were taken on-board to eliminate contamination. Upon recovery the CTD rosette and Niskin sampling bottles were thoroughly rinsed with fresh water on the deck. The Niskin bottles were removed from the sampling frame and transported to a CTD control lab isolated from the deck area where they were mounted on wall brackets for further processing. Prior to sub-sampling the exterior of the Niskin bottles and sampling nozzle were rinsed with a sodium hypochlorite solution (10% commercial bleach) followed by ultrapure water (18.2 MΩ cm⁻¹). Workbench area on-board was covered with aluminium foil and rinsed with a 20% commercial bleach solution, followed by ultrapure water. The foil was replaced after each sampling event.

Sub-sampling bottles were 2 L LDPE bottles that had been thoroughly cleaned beforehand with a 10% HCl solution (24 h soaking), 20% commercial bleach solution (24 h soaking) and subsequently rinsed three times with ultrapure water. Individual sub-sampling bottles were stored in double plastic bags in locked plastic crates that had been washed beforehand with a 20% commercial bleach solution and rinsed with distilled water. The neck diameter of the LDPE sub-sampling bottles was selected such that it formed a perfect seal with the base of the sampling nozzle of the Niskin bottle. The interior of the sub-sampling bottle was therefore

not exposed to the atmosphere during sampling aside from the few seconds it took to remove the cap and dock with the Niskin Nozzle. Each sub-sample bottle was rinsed three times with sample water and then filled to a 1.5 L graduation mark. The 2 L LDPE sample bottles were immediately capped and placed back in double plastic bags and frozen on board at −20 °C to eliminate contamination risks associated with filtering on the research vessel. Sample blanks were taken periodically alongside field samples by filling clean polypropylene bottles with ultrapure water and freezing. All subsequent sample processing took place in a sterile environment at a molecular biology institute ashore.

**eDNA sample filtration.** Upon return to a sterile laboratory samples were stored at −20 °C in a separate wing physically isolated from the molecular laboratory workspace. Samples were subsequently processed in a wet-lab of the same wing. Samples were defrosted at room temperature. Prior to removing the cap, the exterior of the sample bottles were thoroughly washed with tap water (3 min each), followed by a sodium hypochlorite solution and ultrapure water. Immediately prior to filtering, the water samples were homogenised by vigorous shaking for a period of 2 min. Samples were filtered using a peristaltic pump (40 rpm) and particulate material harvested on a 0.2 μm Sterivex filter (Millipore; SVGP01050). Sterivex filters were chosen because they are encapsulated in a plastic cartridge eliminating contact of the filter and sample material with the atmosphere. Residual sample volume in the Sterivex cartridge was evacuated by expelling from the cartridge with a single-use sterile 50 mL luer-lock syringe (Braun; 4617509 F). The female luer-lock inlet of the Sterivex filter was sealed with a male Luer integral lock ring plug (Cole-Parmer; 30800-30). The exterior of the cartridge was wiped with a 10% sodium hypochlorite solution, followed by ultrapure and isolated in a sterile 50 mL Falcon tube. The 50 mL Falcon tubes were stored at -80 °C. Operational field blanks were processed in an identical manner. In between samples, peristaltic pump tubing was rinsed with 1 L of sodium hypochlorite solution (10% commercial bleach solution) and 2 L of ultrapure water.

**DNA extraction.** DNA was extracted from the Sterivex cartidges (Merck Millipore; #SVGP01050) using a modified protocol of the Qiagen DNeasy Blood and Tissue Kit (Qiagen; #69504). All bench space, pipettes and instruments associated with the extraction were cleaned before use with 70% ethanol, followed by RNase away (Qiagen #19101).

Sterivex cartridges were removed from the freezer and allowed to defrost at room temperature for 20 min. A final check on residual volume was performed by expunging air through the Sterivex cartridge with a 50 mL sterile syringe. The male nipple of the Sterivex cartridge was flame sealed and extraction reagents were added directly inside of the cartridge using sterile filter pipette tips. Extraction proceeded according to the manufactuer's instructions with the following modifications. A volume of 720 μL of buffer ATL and 80 μL of proteinase K was added directly to the interior of the Sterivex cartridge. The female inlet was capped with a male luer-lock cap. The Sterivex cartridges were then placed in a rotary spinner and incubated at 56 °C for 2 h. The cartridges were rotated 90° around their central axis every 30 min to ensure even coverage of the filter roll with extraction solution. The lysis soluition was removed from the Sterivex cartridge using a sterile 3 mL luer-lock syringe and transferred to sterile DNase-free 2 mL Eppendorf tubes. The Eppendorfs were pulse vortexed (10 s) and spun down in a mini-centrifuge. Subsequently, 600 μL of extraction solution was transferred to a clean 2 mL Eppendorf tube, followed by 600 μL of buffer AL solution. These Eppendorf tubes were pulse vortexed (10 s) to mix and centrifuged. Taking each Eppendorf in turn, 600 μL of ethanol was added, followed by pulse-vortexing and centrifugation. This extraction solution was added to the Qiagen DNEasy Blood and Tissue kit spin columns in 3 × 600 μL aliquots. The spin column procedure followed the manufacturers instructions.

DNA was eluted from the spin column in 120 μL of nuclease-free water following an incubation period of 5 min at 37 °C. The flow-through from the first elution step was pipetted back onto the column for a second elution under identical conditions. DNA extracts were frozen at −20 °C until further processing. For each batch of extractions an extraction control was included to test for laboratory and reagent contamination. All extraction reagents were added to a sterile Sterivex cartridge and extraction carried out in an identical manner alongside field samples. DNA concentrations in the extracts were measured with a Qubit Fluorometer and Qubit dsDNA HS assay (Q32854), following the manufacturers instructions. DNA concentrations in 1.5 L bottom water samples ranged from 0.16 to 2.3 ng μL⁻¹ (median = 1.17 ng μL⁻¹, mean = 1.23 ± 0.48 ng μL⁻¹).

**qPCR amplification of Gadus morhua eDNA.** Quantitative real-time PCR of Atlantic cod eDNA was carried out using a commercially available *Gadus morhua* speciation assay (Techne, TKIT06035). An Applied Biosystems StepOnePLUS real-time PCR platform was used. Each PCR reaction mixture of 20 μL contained 5 μL of DNA template, 10 μL 2 × qPCR Mastermix (Techne, TKITMM01), 1 μL *Gadus morhua* primer/probe (TaqMan hydrolysis probe), 1 μL Internal Extraction Control primer/probe (TaqMan hydrolysis probe) and 3 μL of nuclease-free water. Primer and probe sequences are deemed proprietary (Techne) but were designed to target the mitochondrial D-loop region of Atlantic cod. PCR reactions were performed under thermocycler conditions of 2 min at 95 °C and 50

cycles of 10 s at 95 °C and 60 s at 60 °C. Fluorogenic data was collected through the FAM and VIC channels.

DNA extracts were not diluted for inhibition checks due to the potentially low yield of target DNA in field samples. Inhibition was tested for in all sample and blank PCR reactions using multiplex PCR and an internal control template (sequence: proprietary-Techne) and primer-probe (sequence: proprietary-Techne) supplied with the Techne Atlantic cod speciation kit (TKIT06035). A volume of 5 μL control template was added to DNA extracts. A comparison of Cq values between samples with internal control template and nuclease-free water with internal control template was used to diagnose inhibition, whereby a Cq shift ≥ 3 cycles was considered as inhibitory[36,37]. Inhibition was not detected in any of the ocean bottom water field samples.

**Negative controls and blanks**. The qPCR assays were set up to include six no template controls (NTC), three contained nuclease-free water in place of template DNA (NTC), and three contained nuclease-free water + internal control template (NTC + IEC). All six NTC wells were used as negative controls for the amplification of G. morhua, and NTC-IEC was used to confirm amplification of the internal control used for comparison with field samples to test for inhibition. None of the NTC samples amplified for G. morhua template. Positive amplification controls were included from the kit-supplied positive control template. Extraction blanks were included with each batch of DNA extraction to control reagent contamination. Extractions were carried out in an identical manner and simultaneously with sample extractions, substituting a sample filter for an unused sterivex filter cartridge. Field blanks were comprised of 1.5 L of ultrapure water collected alongside ocean samples on board the research vessel. Neither extraction blanks nor field sample blanks exhibited amplification with the G. morhua qPCR primers.

**Quantification of copy numbers**. Standard quantitation curves were constructed using a synthetically generated positive control template (kit-supplied) identical in sequence to the mitochondrial target region. Standard curves were generated by serial dilution of the positive control template to create a concentration series of 2, 20, 200, 2000, 20000, 200000 copies μL$^{-1}$. A 10-point concentration series, with ten technical replicates at each concentration was used to estimate an $R^2$ value of 0.99 and PCR efficiency of 81% (Supplementary data 1: MIQE checklist) for the qPCR assay.

**LOD and LOQ of qPCR assay**. LOD and LOQ were determined from an 10-point concentration series, with ten technical replicates at each concentration. LOD was defined as the lowest concentration at which 95% of the technical replicates exhibited positive amplification. LOQ was determined at the lowest concentration at which the relative standard deviation of back-calculated concentrations was <35% (Supplementary Methods). LOD and LOQ were 3 and 20 copies per reaction. Copies per litre were calculated according to the equation: Copies L$^{-1}$ = Copies per reaction × $[(E_{vol}/R_{vol})/S_{vol}]$, where $E_{vol}$ and $R_{vol}$ are the extraction volume and PCR reaction volume (μL) and $S_{vol}$ is the filtered sample volume (L). According to this equation and the methodological set-up LOD and LOQ were 48 and 320 copies L$^{-1}$.

For each field sample, operational sampling blank and laboratory extraction blank, four technical replicates were amplified. Positive amplifications were classified as those samples in which at least ¾ technical replicates amplified above the LOQ. Averages were calculated from these technical replicates and analytical error expressed as ± 1 s.d. (Supplementary Table 4). None of the extraction blanks or operational sampling blanks exhibited any amplification.

**Specificity of qPCR primers to Atlantic cod**. Specificity of Techne primers to Atlantic cod (G. morhua) is based on in-silico analysis of the mitochondrial D-loop region. Additionally we confirmed specificity from the analysis of tissue samples of other Gadidae species collected from the survey area (Table 1). Each species was carefully dissected under sterile conditions and muscle tissue stored in a sterile Falcon tube. DNA was extracted from tissue samples using DNeasy Blood & Tissue (Qiagen) following the manufacturers instructions. qPCR assays were carried out as described above, substituting filter DNA extracts for tissue DNA extracts. Only Atlantic cod tissue samples exhibited amplification with the Techne G. morhua speciation kit.

**Aquarium tests**. Prior to field sampling, the DNA extraction protocol and qPCR assay was tested on 1.5 L water samples collected from the Faroese National aquarium (Føroya Sjósavn). Aquarium tank water was collected from three separate display tanks, each containing Atlantic cod. Sample filtration, DNA extraction and qPCR amplification were carried out as described above. A sample blank of ultrapure water was taken simultaneously. Positive amplification of Atlantic cod eDNA from tank water was observed from all three tanks and the sample blank was negative (Supplementary Fig. 1).

**Statistics and reproducibility**. All statistical analyses were conducted in the R environment[38]. Spatial autocorrelation can invalidate the variable independence assumption of linear regression models[39]. Prior to linear regression Moran's I[40]

was used to test for spatial autocorrelation amongst variables, and subsequent to regression analysis on model residuals[41]. Weight values for Moran's I was calculated using an inverse distance matrix method based on the latitude and longitude of station data, or averages of region data. The null hypothesis of Moran's I (H$_0$) is that spatial autocorrelation does not exist. Z-scores from Moran's I were calculated according to Eq. 1., where I = observed Moran's I, E[I] = expected Moran's I, and SE[I] is the standard deviation of E[I]. Statistical tests for Moran's I was tested at a 95% confidence level (α = 0.05). Spatial autocorrelation calculations were performed using the APE package in R[42].

$$Z(I) = \frac{I - E_{(I)}}{S_{E(I)}}. \tag{1}$$

The correlation between total Atlantic cod biomass per region vs total Atlantic cod DNA copies per region and Catch Per Unit Effort (kg h$^{-1}$) and average Atlantic cod DNA copies per region were evaluated using Type I (OLS) and Type II (RMA) regressions. Type II regressions are preferred when the x variable cannot be assumed to be free of variance[43]. Atlantic cod biomass data was log$_{10}$-transformed to satisfy normality assumptions required for Type I and Type II linear regression. Normality was tested using a Shapiro-Wilk normality test at a significance level 95% (α = 0.05). An ordinary least squares (OLS) method was used for Type I regression. In the Type II regression a Ranged Major Axis (RMA) method was used. Bivariate normality was checked using a Shapiro-Wilk normality test (α = 0.05) to satisfy the assumptions of RMA Type II regression. Statistical significance of regression coefficients were assessed from parameteric p- values (two tailed) at a significance level of (α = 0.05) Regression coefficients A permutation test (99 permutations) was used to determine the significance (α = 0.05) of the slopes and correlation coefficient of Type I-OLS and Type II-RMA regressions. Type I and type II regressions were performed using the R-package lmodel2[44].

Association of presence–absence data was calculated from mean square contingency coefficient[45], commonly referred to as Φ. Trawl detection and eDNA detection rates were treated as binary variables (1 = presence and 0 = absence). The two detection methods were compared in a 2 × 2 contingency table, where a = presence–presence, b = presence–absence, c = absence–presence, d = absence–absence. The phi coefficient is analytically equivalent to the Pearson's product moment correlation (Eq. 2) and can take values from −1 to 1

$$r = \frac{ad - bc}{\sqrt{(a + b)(c + d)(a + c)(b + d)}}. \tag{2}$$

**Reporting summary**. Further information on research design is available in the Nature Research Reporting Summary linked to this article.

## Data availability
The data that support the findings of this study are available from the corresponding author upon reasonable request.

## Code availability
The data code that support the findings of this study are available from the corresponding author upon reasonable request.

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

## Acknowledgements

The authors thank the Fisheries Research Fund of the Faroe Islands (Fiskivinnuroyndir) for financial support for the project COD-e-DNA (2017–2019). Additionally we thank the captain and crew of Magnus Heinason for facilitating sample collection. The authors further acknowledge the assistance of Sólva Eliasen and Dagunn Clementsen in graphic preparation and Hannapouli Olsen for specimen collection. We also thank the Faroese National aquarium for providing access to samples from exhibit tanks and Maria Nielsdottir for assistance with proof-reading applications for funding and earlier manuscript versions.

## Author contributions

I.S. conceived and designed the experiments. I.S., M.J., R.K., P.V., performed sample collection and I.S. performed all laboratory work. P.S. contributed historical trawl data. I.S. performed data analysis and I.S. wrote the paper.

## Competing interests

The authors declare no competing interests.
