## [Peer Review File · Communications Biology]

Reviewers' comments:

Reviewer #1 (Remarks to the Author):

The paper describes a study comparing the abundance of Atlantic cod eDNA in water with the abundance of the same species caught in survey hauls around the Faroe islands in the Northeast Atlantic. The study is timely and novel. Although aquatic eDNA is now becoming increasingly used to monitor the presence or absence of species, it is typically not used to quantitatively measure abundance. This study strengthens the evidence that eDNA can reflect abundance, and therefore may have potential to inform or replace traditional trawl surveys in future years. This is a study that demonstrates strong attention to detail when performing the qPCR experiment, including uses of appropriate blanks, a dilution series that enable the limits of quantification and detection to be estimated, the use of four technical repeats of the qPCR for each sample, and some useful tests of inhibition.

However, there were two key outstanding questions that the manuscript failed to address.

Firstly, I was concerned about assay cross-species amplification. This is a really important factor that is often wilfully overlooked. Here the authors report the use of the proprietary qPCR assay from Techne, which they note is based on the mtDNA control region. But, how certain can the authors be that they have not amplified other gadoid species, of which there are many in the region. This is of note because the assay notes state "The kit is designed to specifically detect Cod species that are relevant to the food industry and to give negative detection on other possible meat species." This text hardly fills me with confidence that the test really is specific to Atlantic cod.

Second, I wondered why the authors did not attempt to compare the number of copies of eDNA in each sample with the actual number for the corresponding trawl? Surely this is the level of granularity that is required if eDNA will prove to be a useful tool?

Smaller points:

Line 27. I would expect to see a brief description of what eDNA is, for the interested by non-specialist reader.

Line 83. G. morhua

Line 91. Not clear what is meant by "incidence"

Line 106. Are you sure you have the right units here... copies per region. Surely that should be copies per unit volume of seawater or DNA concentrate.

Line 247. G. morhua.

Figure 1b. Need to explain what the error bars etc. (as it standard for boxplots).

Figure 2f. No mention of what error bars represent.

Reviewer #2 (Remarks to the Author):

This work using Environmental DNA (eDNA) showed that eDNA copy numbers in bottom water quantitatively map the regional distribution of Catch per unit effort (CPUE) for Atlantic cod. It was known the positive relationship between eDNA copy and biomass, but scarcely understood in ocean environments. So, this finding would be interesting for marine as well as freshwater ecologists and fishery researchers. The paper was well written, but the technical methods and the descriptions are not sufficient to publish. However, I had some comments on the MS as follows.

Please refer MIQE checklist (<http://www.rdml.org/miqe.php>) to describe qPCR methods, the current version missed many descriptions such as R2, PCR efficiency...

I am wondering the testing species specificity of the commercial primer set; Techne, TKIT06035.

Usually, species-specific primer was tested by Primer-BLAST and in-vivo test using the DNA extracted by the tissue samples, including the related species. Please refer the qPCR primer guideline described in Goldberg et al. 2017 <https://besjournals.onlinelibrary.wiley.com/doi/abs/10.1111/2041-210X.12595>

Reviewer #3 (Remarks to the Author):

The study aims are interesting, particularly the field-based assessment of eDNA derived abundances. However, the results lack statistical support for several key statements with no statistical analyses details provided in the methods. Methods need to provide clear descriptions of the molecular tools used. Large amount of literature ignored pertaining to abundances derived from eDNA survey methods. The study, as presently presented, is highly specialized as the focus is on a single species and a specific method for quantifying abundances which may not reach a wide audience.

Please provide results within the abstract. There is currently no descriptive description of the experimental design or statistical support for the conclusions stated

36-37: There have been efforts to quantify fish via eDNA from several papers, primarily out of labs originating in Japan. A couple of key papers show experimental mesocosm can be quantified. Field based quantifications are still an issue.

61-64: how stable are these ecoregions? How has fisheries management influenced these stocks overtime? What are the limitations of using trawling for quantifying fish abundances? These comments should be addressed in the introduction

99-102: As each measure should represent the abundance for each sample, it is not appropriate to sum the data to reach a single abundance measure. Calculating the mean or median would be suitable, however a greater test of the data would include the data points with replicate as a covariate using a statistical framework.

102-104: Please provide statistical support for the biomass-eDNA signal relationship

118-123: Since the fish abundance and eDNA derived abundances are interchangeable response variables for type I regressions a type II regression should be used to assess the trends. See:

Doi, Hideyuki, et al. "Use of droplet digital PCR for estimation of fish abundance and biomass in environmental DNA surveys." *PloS one* 10.3 (2015): e0122763.

125-134: As there is a spatial component to the sampling. There needs to be some consideration in the methods and the discussion for the autocorrelation of the sampling design and analysis that can lead to violations of independence for many statistical methods, including type I linear regressions.

140-141: Limiting the literature comparison to studies that focus on CPUE limits the application of the findings. Again, a large section of the literature is being ignored that would be relevant to address

226-234: provide the primer/probe sequences and the relevant descriptive statistics including lengths,

melting temperature, gc content. Also provide the thermo cycler protocols and the sequence of the synthetic control along with the sequence length

Discussion: eDNA data is generally not site specific in aquatic systems due movement of the physical environment and is sensitive to the local abiotic conditions.

Deiner, K., Bik, H.M., Mächler, E., Seymour, M., Lacoursière-Roussel, A., Altermatt, F., Creer, S., Bista, I., Lodge, D.M., De Vere, N. and Pfrender, M.E., 2017. Environmental DNA metabarcoding: Transforming how we survey animal and plant communities. *Molecular ecology*, 26(21), pp.5872-5895.

Seymour, M., Durance, I., Cosby, B.J., Ransom-Jones, E., Deiner, K., Ormerod, S.J., Colbourne, J.K., Wilgar, G., Carvalho, G.R., de Bruyn, M. and Edwards, F., 2018. Acidity promotes degradation of multi-species environmental DNA in lotic mesocosms. *Communications biology*, 1(1), p.4.

Methods: there are no statistical methods included. Please provide a full description of the analyses performed to support the results/conclusion drawn from the data.

Reviewer #1 (Remarks to the Author):

R1 – Q/C – 1: The paper describes a study comparing the abundance of Atlantic cod eDNA in water with the abundance of the same species caught in survey hauls around the Faroe islands in the Northeast Atlantic. The study is timely and novel. Although aquatic eDNA is now becoming increasingly used to monitor the presence or absence of species, it is typically not used to quantitatively measure abundance. This study strengthens the evidence that eDNA can reflect abundance, and therefore may have potential to inform or replace traditional trawl surveys in future years.

R1 – R – 1: We would like to thank the reviewer for their positive attitude and appreciation of our study. We agree that in order to advance the field of eDNA for ecosystem assessment, particularly with respect to fisheries, studies that compare quantitative eDNA surveys with traditional survey methods in the marine environment are required. That was indeed the motivation for the present study.

R1 – Q/C – 2: This study has demonstrated strong attention to detail when performing the qPCR experiment, including uses of appropriate blanks, a dilution series that enable the limits of quantification and detection to be estimated, the use of four technical repeats of the qPCR for each sample, and some useful tests of inhibition.

R1 – R – 2: Again we would like to thank the reviewer for these compliments. We went to extraordinary lengths to ensure that our qPCR assay was as reliable as possible and that all the necessary controls and checks with respect to collection and analyzing field samples were addressed. We are therefore extremely confident in our results and very excited by the prospect that eDNA surveys can be as informative as traditional trawl surveys in documenting the regional distribution of Atlantic cod biomass around the Faroe Islands.

R1 – Q/C – 3: However, there were two key outstanding questions that the manuscript failed to address. Firstly, I was concerned about assay cross-species amplification. This is a really important factor that is often wilfully overlooked. Here the authors report the use of the proprietary qPCR assay from Techne, which they note is based on the mtDNA control region. But, how certain can the authors be that they have not amplified other gadoid species, of which there are many in the region. This is of note because the assay notes state “The kit is designed to specifically detect Cod species that are relevant to the food industry and to give negative detection on other possible meat species.” This text hardly fills me with confidence that the test really is specific to Atlantic cod.

R1 – R – 3: Following discussions with the supplier regarding their primer for *G.morhua* we felt assured that it was species-specific, particularly because this primer was developed for detecting food-fraud which is bound by strong legal legislation. However, we fully appreciate the need to respond to the reviewers concerns about specificity, which were also shared by one other reviewer. In order to address this we reanalyzed the trawl data from the five-week survey period in 2018 to identify all the species of fish present and classified those belonging to the Gadidae family. These are presented in Table 1 in order of decreasing biomass.

Table 1:

Common Name	Scientific Name	Total Biomass (x 10 ³ kg)	Ratio with cod	Tissue Type	Ct
Atlantic Cod	Gadus morhua	18.8	1.00	Muscle	18.8 ± 0.56
Haddock	Melanogrammus aeglefinus	12.9	0.69	Muscle	Undetermined
Saithe	Pollachius virens	1.79	0.10	Muscle	Undetermined
Norway Pout	Trisopterus esmarki	0.88	0.05	Muscle	Undetermined
Whiting	Merlangius merlangus	0.58	0.03	Muscle	Undetermined
Blue Whiting	Micromestisius poutassi	0.31	0.02	Fin	Undetermined
Poor Cod	Trisopterus Minutus	0.03	0.002	No sample	No sample

Silvery Pout	Gadiculus Argenteus	0.01	0.001	Muscle	Undetermined
----------------------------	------	-------	--------	--------------

Following this data reanalysis we organized the collection of these specimens from our August 2019 survey (partly explaining the delay in the response this review) to obtain fresh tissue samples. We extracted DNA from 20 mg of muscle tissue (or a fin clipping in the case of Blue Whiting) and extracted it using the same extraction and qPCR protocol as applied to our field samples. From this we are able to demonstrate that the primer used in this study is specific to Atlantic Cod and does not amplify DNA from any of the closely related *Gadidae* species present in the survey area (Table 1). Unfortunately we could not perform the test on Poor Cod as we were unable to collect a field sample in the 2019 survey due to its scarcity. In fact this species is only present in three of the eight sampling regions in significant quantities (Table 2). In overall study area it has a biomass three orders of magnitude lower than that of Atlantic Cod (Table 1).

Region	Atlantic Cod	Haddock	Saithe	Norway Pout	Whiting	Blue Whiting	Poor Cod	Silvery Pout
North	2791	916	0.00	0.00	9.45	0.00	0.70	0.00
West	14405	838	32.3	0.54	0.00	0.00	0.00	0.00
South	261	384	1141	1.74	0.00	0.00	0.00	0.00
East Coast	734	719	73.3	0.31	124	0.00	0.00	0.00
East Shelf	21.2	582	167	635	412	0.00	0.00	10.7
East Deep	2.96	61.6	216	206	14.0	292	0.00	0.00
Bank Edge	18.8	170	48.0	1.87	1.21	20.9	7.14	0.37
Bank Central	596	9191	109	35.4	20.0	0.00	20.8	0.00

We have now included these tests in the methods (lines 464-473) and results (Table 1, lines 82-97) sections of the manuscript and hope that they satisfactorily address the reviewers concerns regarding the specificity of the *G. morhua* primer.

R1 – Q/C – 4: Second, I wondered why the authors did not attempt to compare the number of copies of eDNA in each sample with the actual number for the corresponding trawl? Surely this is the level of granularity that is required if eDNA will prove to be a useful tool?

R1 – R – 4: We actually did make this comparison and it was presented in the original manuscript as Supplementary Figures 9 and 10 and in Lines 147-157 of the main manuscript. We are of the view that in marine systems station by station granularity is not required for eDNA to be a useful tool. Trawl data itself is highly variable from station to station, particularly in the case of aggregating species like Atlantic Cod. That is why catch data is often integrated across larger regions from numerous trawls. Furthermore, physical mechanisms that move, and possibly dilute, DNA away from source organisms, are of significance in larger oceanic settings. That is why the rationale and hypothesis of our study was to test the relationships over broader regional scales that exhibit a known natural biomass gradient based on historical trawl data. We feel like it is an important finding, that we perhaps have not focused on enough, that at fine-scale granularity we observe robust statistical relationships between presence-absence of the two techniques, but not quantitative relationships. However, at a broader spatial scale we do observe quantitative relationships. That is why we conclude that eDNA surveys can be useful for identifying biomass patterns at regional scales in the ocean, and possibly over time in a defined region, but not in terms of quantifying the number of fish at a specific location. We feel identifying these positive relationships, and highlighting that they occur as a function of scale is an important finding and consideration for the application of eDNA surveys to marine fisheries.

We have moved the station-by-station comparison out of the supplementary material and into the main body of the manuscript (New Figure 3d). We have also added an extra paragraph in the discussion to address these considerations (lines 216 – 232).

R1 – Q/C – 5- Line 27. I would expect to see a brief description of what eDNA is, for the interested by non-specialist reader.

R1 – R – 5 – A brief description of eDNA has been added to the revised introduction: Lines 35-39

R1 – Q/C – 6 - Line 83. *G. morhua*

R1 – R – 6 – We have changed *G. morhua* to Atlantic cod throughout the manuscript.

R1 – Q/C – 7 Line 91. Not clear what is meant by “incidence”

R1 – R – 7 – Removed.

R1 – Q/C – 8 - Line 106. Are you sure you have the right units here... copies per region. Surely that should be copies per unit volume of seawater or DNA concentrate.

R1 – R – 8 – Units corrected.

R1 – Q/C – 9 - Line 247. *G. morhua*.

R1 – R – 9 – We have changed *G. morhua* to Atlantic cod throughout the manuscript.

R1 – Q/C – 10 - Figure 1b. Need to explain what the error bars etc. (as it standard for boxplots)..

R1 – R – 10 – An explanation of box plot characteristics has been added to Figure 1 caption.

R1 – Q/C – 11 Figure 2f. No mention of what error bars represent

R1 – R – 11 – Figure 2f is now Figure 3b. A description of error bars added to figure caption.

Reviewer #2 (Remarks to the Author):

R1 – Q/C – 1: This work using Environmental DNA (eDNA) showed that eDNA copy numbers in bottom water quantitatively map the regional distribution of Catch per unit effort (CPUE) for Atlantic cod. It was known the positive relationship between eDNA copy and biomass, but scarily understanding in ocean environments. So, this finding would be interesting for marine as well as freshwater ecologists and fishery researchers. The paper was well written, but the technical methods and the descriptions are not sufficient to publish, However, I had some comments on the MS as follows.

R1 – R – 1 – We would like to thank the reviewer for these positive words and express our gratitude in them taking the time to review our manuscript.

R1 – Q/C – 2 – Please refer MIQE checklist (<http://www.rdml.org/miqe.php>) to describe qPCR methods, the current version missed many descriptions such as R2, PCR efficiency...

R1 – R – 2 – We have added a MIQE checklist as a supplementary information file. R2 and PCR efficiency have been added to the methods section. Lines 437-443

R1 – Q/C – 3 – I am wondering the testing species specificity of the commercial primer set; Techne, TKIT06035. Usually, species-specific primer was tested by Primer-BLAST and in-vivo test using the DNA extracted by the tissue samples, including the related species. Please refer the qPCR primer guideline described in Goldberg et al. 2017

R1 – R – 3 – Following discussions with the supplier regarding their primer for *G.morhua* we felt assured that it was species-specific, particularly because this primer was developed for detecting food-fraud which is bound by strong legal legislation. However, we fully appreciate the need to respond to the reviewers concerns about specificity, which were also shared by one other reviewer. In order to address this we reanalyzed the trawl data from the five-week survey period in 2018 to identify all the species of fish present and classified those belonging to the Gadidae family. These are presented in Table 1 in order of decreasing biomass.

Table 1:

Common Name	Scientific Name	Total Biomass (x 10 ³ kg)	Ratio with cod	Tissue Type	Ct
Atlantic Cod	Gadus morhua	18.8	1.00	Muscle	18.8 ± 0.56
Haddock	Melanogrammus aeglefinus	12.9	0.69	Muscle	Undetermined
Saithe	Pollachius virens	1.79	0.10	Muscle	Undetermined
Norway Pout	Trisopterus esmarki	0.88	0.05	Muscle	Undetermined
Whiting	Merlangius merlangus	0.58	0.03	Muscle	Undetermined
Blue Whiting	Micromestisius poutassi	0.31	0.02	Fin	Undetermined
Poor Cod	Trisopterus Minutus	0.03	0.002	No sample	No sample
Silvery Pout	Gadiculus Argenteus	0.01	0.001	Muscle	Undetermined

Following this data reanalysis we organized the collection of these specimens from our August 2019 survey (partly explaining the delay in the response this review) to obtain fresh tissue samples. We extracted DNA from 20 mg of muscle tissue (or a fin clipping in the case of Blue Whiting) and extracted it using the same extraction and qPCR protocol as applied to our field samples. From this we are able to demonstrate that the primer used in this study is specific to Atlantic Cod and does not amplify DNA from any of the closely related *Gadidae* species present in the survey area (Table 1). Unfortunately we could not perform the test on Poor Cod as were unable to collect a field sample in the 2019 survey due to it's scarcity. In fact this species is only present in three of the eight sampling regions in significant quantities (Table 2). In overall study area it has a biomass three orders of magnitude lower than that of Atlantic Cod (Table 1).

Region	Atlantic Cod	Haddock	Saithe	Norway Pout	Whiting	Blue Whiting	Poor Cod	Silvery Pout
North	2791	916	0.00	0.00	9.45	0.00	0.70	0.00
West	14405	838	32.3	0.54	0.00	0.00	0.00	0.00
South	261	384	1141	1.74	0.00	0.00	0.00	0.00
East Coast	734	719	73.3	0.31	124	0.00	0.00	0.00
East Shelf	21.2	582	167	635	412	0.00	0.00	10.7
East Deep	2.96	61.6	216	206	14.0	292	0.00	0.00
Bank Edge	18.8	170	48.0	1.87	1.21	20.9	7.14	0.37
Bank Central	596	9191	109	35.4	20.0	0.00	20.8	0.00

We have now included these tests in the methods (lines 464-473) and results (Table 1, lines 82-97) sections of the manuscript and hope that they satisfactorily address the reviewers concerns regarding the specificity of the *G. morhua* primer.

Reviewer #3 (Remarks to the Author):

R3 – Q/C – 1 - The study aims are interesting, particularly the field-based assessment of eDNA derived abundances. However, the results lack statistical support for several key statements with no statistical analyses details provided in the methods. Methods need to provide clear descriptions of the molecular tools used. Large amount of literature ignored pertaining to abundances derived from eDNA survey methods. The study, as presently presented, is highly specialized as the focus is on a single species and a specific method for quantifying abundances which may not reach a wide audience.

R3 – R – 1 – We would like to thank the reviewer for expressing their interest in our study aims and for their comprehensive review. The comments above are a summary and revisited in the detailed comments below. We have addressed the statistic

R3 – Q/C – 2 - Please provide results within the abstract. There is currently no descriptive description of the experimental design or statistical support for the conclusions stated

R3 – R – 2 - Results have been added to the abstract at lines 17-22. The experimental design is now described at the end of the introduction (lines 73-78). Statistical support for the stated conclusions have also been added to the abstract lines 17-22.

R3 – Q/C – 3 - There have been efforts to quantify fish via eDNA from several papers, primarily out of labs originating in Japan. A couple of key papers show experimental mesocosm can be quantified. Field based quantifications are still an issue.

R3 – Q/C – 3 – As part of the revision process we have made significant changes to the introduction. The new introduction includes the important results mentioned above by the reviewer.

R3 – Q/C – 4 - how stable are these ecoregions? How has fisheries management influenced these stocks overtime? What are the limitations of using trawling for quantifying fish abundances? These comments should be addressed in the introduction.

R3 – Q/C – 4 – The ecoregions are quite stable with respect to each other, although there is some interannual variability that in part can be attributed to overfishing. We have now included an additional figure in the paper (Figure 1d) to show how the biomass of cod within these ecoregions has changed over time. We have added some additional information to the discussion concerning the impact of fisheries management on the Faroese cod stock (lines 68-72). We have added a section to the introduction to address the limitations of trawling (lines 30-34).

R3 – Q/C – 5 - As each measure should represent the abundance for each sample, it is not appropriate to sum the data to reach a single abundance measure. Calculating the mean or median would be suitable, however a greater test of the data would include the data points with replicate as a covariate using a statistical framework.

R3 – R – 5 - We summed the data as we wanted to simply address the hypothesis that the total biomass of fish in a defined region is correlated with the total amount of DNA originating from that fish. Fish catch data is often treated in this way, but commonly normalized to sampling effort (e.g. Catch Per Unit Effort (CPUE) and presented as kg / hr. Since our experimental design had a paired eDNA-trawl sample design, e.g. one water sample corresponding to one trawl we considered it was informative and practical to treat the eDNA data in a similar fashion. Following the reviewers suggestion we now explicitly compare CPUE (kg / hr) with DNA copies per Litre in the same region and present this as a new figure (Figure 3c). Additionally we include a comparison of the station-by station relationship between CPUE and DNA concentrations (Figure 3d).

R3 – Q/C – 6 -Please provide statistical support for the biomass-eDNA signal relationship

R3 – R – 6 – We have now provided further statistical support for the biomass-eDNA signal relationship in the results section. Lines 180-190.

R3 – Q/C – 7 -Since the fish abundance and eDNA derived abundances are interchangeable response variables for type I regressions a type II regression should be used to assess the trends. See:

Doi, Hideyuki, et al. "Use of droplet digital PCR for estimation of fish abundance and biomass in environmental DNA surveys." *PloS one* 10.3 (2015): e0122763.

R3 – R – 7 - It is our understanding that the choice of using type I and type II regressions is based on whether both the explanatory (independent) and response (dependent) variable are random, i.e. not controlled by the researcher. That is because in type I regressions, which typically regress a dependent variable (Y) on an independent variable (X), the line of best fit is found by minimizing the residuals (Ordinary Least Squares (OLS) method) on the y-offsets. As such a type I regression assumes that there is little or no measurement error in the independent variable (X), and that most, or all of the error on the regression is associated with the dependent variable (Y).

In field datasets, such as ours, and others (e.g. Knudsen et al. 2019) it is slightly more obscure because we are not explicitly controlling the number of fish caught in the trawls rather taking it as an independent biomass estimate and assessing whether there is a measurable and dependent response of DNA concentrations. However, in the case of field samples it is probably statistically correct to use a Type II regression as technically we are comparing two measurements, the biomass of cod in the trawl and the concentration of cod-derived DNA in water samples. In this case a type I regression using least squares fit could underestimate the slope of the linear regression.

In order to address the concern of the reviewer regarding type I and type II linear regressions, we compared them both and found similar results. After careful consideration we agree with the reviewer that even though other authors have used type I regressions to compare fish biomass and eDNA in the field it is statistically more appropriate to apply a type II regression. We have added Type II regression lines to our biomass-eDNA signals (Figure 3).

In the revised version we have added a description of the statistical methods used for regression (lines 499-514) and the results (lines 180-190).

R3 – Q/C – 8 - As there is a spatial component to the sampling. There needs to be some consideration in the methods and the discussion for the autocorrelation of the sampling design and analysis that can lead to violations of independence for many statistical methods, including type I linear regressions.

R3 – R – 8 – We thank the reviewer for highlighting this; it is something we should have included in the original version. We tested our dataset for spatial autocorrelation using Moran's I. Our data did not exhibit statistically significant spatial autocorrelation and therefore does not violate the assumptions of linear regression. We have included a description of this approach in the statistical methods section (lines 486-498) and results section (lines 176-178).

R3 – Q/C – 9 - Limiting the literature comparison to studies that focus on CPUE limits the application of the findings. Again, a large section of the literature is being ignored that would be relevant to address

R3 – R – 9 – We have expanded the literature included in the manuscript to include those studies that have made comparisons of biomass and eDNA concentrations, for example echo-sounding, visual surveys etc.

R3 – Q/C – 10 -provide the primer/probe sequences and the relevant descriptive statistics including lengths, melting temperature, gc content. Also provide the thermo cycler protocols and the sequence of the synthetic control along with the sequence length

R3 – R – 10 – We have now completed a MIQE document and include it as supplementary information file to fully document the required technical information of the PCR assay.

R3 – Q/C – 11 - Discussion: eDNA data is generally not site specific in aquatic systems due movement of the physical environment and is sensitive to the local abiotic conditions.

R3 – R – 11 – We agree with this statement. We have included a more explicit consideration of this in the revised discussion, including references. (lines 240-243).

R3 – Q/C – 12 - Methods: there are no statistical methods included. Please provide a full description of the analyses performed to support the results/conclusion drawn from the data.

R3 – R – 12 – We have added a section to the methods to describe the statistical analysis. (Lines 484-522).

Reviewers' comments:

Reviewer #1 (Remarks to the Author):

My main concern with the previous version of the manuscript was the absence of evidence for species specificity of the qPCR assay. The authors have now adequately demonstrated the specificity, and the manuscript is much improved. I have a few relatively minor comments that the authors should consider, but overall the authors should be congratulated on this study that will make a very positive contribution to the marine eDNA literature.

Lines 17-22 I recommend avoiding reporting statistics in the abstract, and instead report the key findings in sentences.

Lines 85-90, and elsewhere (e.g. table 1). Ensure the species names are correctly reported. When reporting Latin names only the species name should have the first letter capitalized. When reporting common names, do not capitalize the first letter unless a) starting a sentence, or b) the name has etymology from a location or person. For example, you should write Atlantic cod, blue whiting and poor cod.

Line 94. Perhaps be more circumspect regarding the biomass of poor cod. It is a small species so the catch will depend entirely on the cod end mesh.

Lines 135 to 140. You seem to write the same results twice. Perhaps delete the sentence starting "Trawl and eDNA detection..."

Line 145, and elsewhere. The concept of a negative CPUE seems a bit strange. Maybe rephrase the trawl catches to present or absent, instead of positive and negative.

Lines 169-178. While I can see the benefit of log transformation of the trawl biomass data, this would mainly be to reduce the incidence of extreme outliers that can unduly influence the outcome of regression analyses. This would not necessarily be to ensure that either the dependent or independent variables are normally distributed. In a legitimate regression analysis both can deviate from normality, as long as the residuals of independent variable in the fitted model are normally distributed. Hence, you can simplify this paragraph by deleting the details of the Shapiro-Wilk test etc.

Line 210. Use "mackerel tuna"

Reviewer #2 (Remarks to the Author):

I carefully checked the revised manuscript, and agree your all responses. I think you work well to revise the manuscript especially the method description and primer specificity.

Reviewer #3 (Remarks to the Author):

Comments to Authors - COMMSBIO-19-0618A

The revision has substantially improved with regards to the methods and results section making the study more discernable. A general minor remark is that there are some statements in the results section that still need statistical support. A major concern is that the primer information, particularly

the primer/probe sequences and the positive control template used are still missing from the main text making the study non-repeatable by any readers and making assessment of the sequences for review purposes also impossible. The authors have provided a miqc checklist for the primers, however this is not a substitute for the primer information themselves. That the primer, probe and template sequences are deemed propriety was not disclosed in the original draft.

Another major issue is that the distribution of the copy numbers is heavily skewed and needs to be transformed to use the statistical test implemented and shown in figure 3d (the main premise of the study). Furthermore, considerations need to be given to how to handle the zero inflation of the data as a result of the large number of zero copy number sites included in the analyses. Some possible solutions from a quick analyses of your data from supplementary table 6. Consider removing the sites with zero values for both methods to avoid zero inflation on such a small data set. Rerunning your stats from supplementary table 6 you have a highly skewed copy number distribution with two clusters for the copies (0s and other values) meaning the presented stat values are violating some key assumptions of the models used. Taking the square root of the copies brought the distribution closer to normal, however there was still evidence of zero inflation from plotting the data and the fitted curve. The cleanest test I could manage was to remove the sites with 0 copies and log transform the copy numbers whereby the correlations were non-significant at $R^2=0.055$ and $p\text{-value} = 0.304$ across 21 sites. You may wish to perform some form of subsampling to include 0 sites into the analysis.

Overall, I do not think your data supports the finding that copy numbers correlated with CPU, but that the copy values are instead randomly associated per site, possibly due to different environmental factors attributed to the collection of each data type (physical collection versus environmental sampling).

Further minor comments

Line 82: please provide the company location

Line 155-158: This statement need statistical support otherwise it is an opinion.

Line 166-167: please provide statistical support for this statement.

Line 187-190 & Figure 3. Panel d looks like a possible violation of homogeneity of variance. The full statistical summary for each of these test should be included as a table

Reviewers' comments:

Reviewer #1 (Remarks to the Author):

R1 – Q/C – 1: My main concern with the previous version of the manuscript was the absence of evidence for species specificity of the qPCR assay. The authors have now adequately demonstrated the specificity, and the manuscript is much improved. I have a few relatively minor comments that the authors should consider, but overall the authors should be congratulated on this study that will make a very positive contribution to the marine eDNA literature.

R1 – R – 1: We would like to thank Reviewer 1 for their positive comments and time invested in reviewing the manuscript. Their suggestions have significantly improved the clarity of our findings.

R1 – Q/C – 2: Lines 17-22 I recommend avoiding reporting statistics in the abstract, and instead report the key findings in sentences.

R1 – R – 2: The statistics were placed in the abstract at the behest of another reviewer. We will allow the editorial staff to make the final decision.

R1 – Q/C – 3: Lines 85-90, and elsewhere (e.g. table 1). Ensure the species names are correctly reported. When reporting Latin names only the species name should have the first letter capitalized. When reporting common names, do not capitalize the first letter unless a) starting a sentence, or b) the name has etymology from a location or person. For example, you should write Atlantic cod, blue whiting and poor cod.

R1 – R – 3: These typos have been corrected.

R1 – Q/C – 4: Line 94. Perhaps be more circumspect regarding the biomass of poor cod. It is a small species so the catch will depend entirely on the cod end mesh.

R1 – R – 4: An interesting point, and we certainly agree that we should be cautious about the selectivity of trawls when comparing with non-selective eDNA data. In the particular case of poor cod, we don't think that size selectivity is a major issue. The mesh-size of our survey trawl is 40mm (Line 290 in methods). We checked the historical trawl data from the site to confirm the size characteristics of poor cod in the study area. It has a mean size of 19.8 ± 0.33 cm, so is adequately sampled by our survey trawl with respect to size.

R1 – Q/C – 5: Lines 135 to 140. You seem to write the same results twice. Perhaps delete the sentence starting "Trawl and eDNA detection..."

R1 – R – 5: This was an editing mistake. It has now been deleted. Thanks again for your careful attention to detail.

R1 – Q/C – 6: Line 145, and elsewhere. The concept of a negative CPUE seems a

but strange. Maybe rephrase the trawl catches to present or absent, instead of positive and negative.

R1 - R - 7: We have changed the terminology to positive detection and negative detection with respect to the various methods.

R1 - Q/C - 7: Lines 169-178. While I can see the benefit of log transformation of the trawl biomass data, this would mainly be to reduce the incidence of extreme outliers that can unduly influence the outcome of regression analyses. This would not necessarily be to ensure that either the dependent or independent variables are normally distributed. In a legitimate regression analysis both can deviate from normality, as long as the residuals of independent variable in the fitted model are normally distributed. Hence, you can simplify this paragraph by deleting the details of the Shapiro-Wilk test etc.

R1 - R - 7: Thanks for this comment. We have specified we were referring to residual normality.

R1 - Q/C - 8: Line 210. Use “mackerel tuna”

R1 - R - 8: OK.

Reviewer #2 (Remarks to the Author):

R2 - Q/C - 1: I carefully checked the revised manuscript, and agree your all responses. I think you work well to revise the manuscript especially the method description and primer specificity.

R2 - R - 1: Thank you.

Reviewer #3 (Remarks to the Author):

Comments to Authors - COMMSBIO-19-0618A

R3- Q/C - 1 - The revision has substantially improved with regards to the methods and results section making the study more discernable. A general minor remark is that there are some statements in the results section that still need statistical support. A major concern is that the primer information, particularly the primer/probe sequences and the positive control template used are still missing from the main text making the study non-repeatable by any readers and making assessment of the sequences for review purposes also impossible. The authors have provided a miqe checklist for the primers, however this is not a substitute for the primer information themselves. That the primer, probe and template sequences are deemed propriety was not disclosed in the original draft.

R3 – R – 1: We would like to thank the reviewer for their positive comments regarding our revisions. We are especially grateful to reviewer 3 and the time they have invested in carefully reviewing our manuscript. Their suggestions led to substantial improvements in the clarity of our results and discussion.

Concerning the primer/probe sequences, it is regrettable that we cannot disclose this information due to propriety reasons. However, we discussed this issue with the editorial board prior to resubmission. We consider that this does not influence the repeatability of the study as the primers can be acquired and similar tests experiments run with the same primers, even if the base sequences are not known.

R3- Q/C – 2 - Another major issue is that the distribution of the copy numbers is heavily skewed and needs to be transformed to use the statistical test implemented and shown in figure 3d (the main premise of the study).

Overall, I do not think your data supports the finding that copy numbers correlated with CPU, but that the copy values are instead randomly associated per site, possibly due to different environmental factors attributed to the collection of each data type (physical collection versus environmental sampling).

R3 – R – 2: Firstly we would like to state that we agree with your statement that Figure 3d does not support a strong (or potentially even significant; see below) correlation between copy numbers and CPUE data on a station-by-station. The main conclusion of our manuscript is that regional variations in copy number map regional biomass indices as measured by standardized trawls.

We moved Figure 3d into the main body of the manuscript in response to reviewers comments. This was a useful suggestion as it helped us clarify the point that in oceanic systems, although good agreement is found at regional scales, it breaks down on a station-by-station basis for concentration data (but is much better for presence-absence statistics). We addressed this in the discussion and attribute it to physical dispersal and decay factors (Discussion lines 216-232 in previous version). In this sense, it does not strongly influence the main conclusions of our manuscript if the station-by-station comparison in Figure 3d is weakly correlated or not statistically correlated (see below).

R3 – Q/C – 3: Furthermore, considerations need to be given to how to handle the zero inflation of the data as a result of the large number of zero copy number sites included in the analyses. Some possible solutions from a quick analyses of your data from supplementary table 6. Consider removing the sites with zero values for both methods to avoid zero inflation on such a small data set. Rerunning your stats from supplementary table 6 you have a highly skewed copy number distribution with two clusters for the copies (0s and other values) meaning the presented stat values are violating some key assumptions of the models used. Taking the square root of the copies brought the distribution closer to normal, however there was still evidence of zero inflation from plotting the data and the fitted curve. The cleanest test I could manage was to remove the

sites with 0 copies and log transform the copy numbers whereby the correlations were non-significant at $R^2=0.055$ and $p\text{-value} = 0.304$ across 21 sites. You may wish to perform some form of subsampling to include 0 sites into the analysis.

R3 – R –3: Firstly we would like to thank you again for the time and effort you have invested in examining our dataset. It is very much appreciated and we feel extremely fortunate to have received such competent and thorough reviews.

We acknowledge your concern that the number of zero copy number data results in a skewed distribution that may violate one of the assumptions of linear regression models. Violating the normality assumption of linear regression of course simply means that the regression coefficient and variance estimates could be biased. However, as mentioned above we actually use the station-by-station comparison to argue that there is little evidence of a correlation between CPUE and copy numbers at this fine scale in the ocean. In this context it is not critical to our arguments if the regression co-efficient of 0.15 is potentially biased since we use it to support our argument there is a weak correlation on a station-by-station basis. Having said that, we followed the most conservative approach of your suggestion by removing all zeros and running the type II regression again, which as you point out shows a non-significant correlation. We added this to the results section (lines 189-191) and discussion (lines 231-236) to further emphasise the poor and uncertain statistical correlation between CPUE and copy number concentrations in the ocean on a station by station basis.

Further minor comments

R3 – Q/C – 4: Line 82: please provide the company location

R3 – R – 4: Added (Line 82).

R3 – Q/C – 5: Line 155-158: This statement need statistical support otherwise it is an opinion.

R3 – R – 5: We have added the regression statistics between the regional ranks as statistical support for the this statement.

R3 – Q/C – 6: Line 166-167: please provide statistical support for this statement.

R3 – R – 6: We have added information about residual variance as statistical support for this statement.

R3 – Q/C – 7: Line 187-190 & Figure 3. Panel d looks like a possible violation of homogeneity of variance. The full statistical summary for each of these test should be included as a table.

R3 - R - 7: We have partially addressed this above. We checked homoscedasticity of this dataset (including zeros) by calculating a scale plot to examine whether or not variance is systematically increasing over the range of fitted values. We found that the data tended more towards homoscedasticity than heteroscedasticity (Figure R3). However, as mentioned above, we do not wish to confuse matters by implying possible bias on a correlation we present as weak.